# In Vivo Preclinical Assessment of the VEGF Targeting Potential of the Newly Synthesized [^52^Mn]Mn-DOTAGA-Bevacizumab Using Experimental Cervix Carcinoma Mouse Model

**DOI:** 10.3390/diagnostics13020236

**Published:** 2023-01-08

**Authors:** Csaba Csikos, Adrienn Vágner, Gábor Nagy, Ibolya Kálmán-Szabó, Judit P. Szabó, Minh Toan Ngo, Zoltán Szoboszlai, Dezső Szikra, Zoárd Tibor Krasznai, György Trencsényi, Ildikó Garai

**Affiliations:** 1Division of Nuclear Medicine and Translational Imaging, Department of Medical Imaging, Faculty of Medicine, University of Debrecen, H-4032 Debrecen, Hungary; 2Gyula Petrányi Doctoral School of Clinical Immunology and Allergology, Faculty of Medicine, University of Debrecen, H-4032 Debrecen, Hungary; 3Scanomed Ltd., H-4032 Debrecen, Hungary; 4Department of Obstetrics and Gynaecology, Faculty of Medicine, University of Debrecen, H-4032 Debrecen, Hungary

**Keywords:** ^52^Mn, bevacizumab, cervix carcinoma, positron emission tomography, VEGF

## Abstract

Among humanized monoclonal antibodies, bevacizumab specifically binds to vascular endothelial growth factor A (VEGF-A). VEGF-A is an overexpressed biomarker in cervix carcinoma and is involved in the development and maintenance of tumor-associated neo-angiogenesis. The non-invasive positron emission tomography using radiolabeled target-specific antibodies (immuno-PET) provides the longitudinal and quantitative assessment of tumor target expression. Due to antibodies having a long-circulating time, radioactive metal ions (e.g., ^52^Mn) with longer half-lives are the best candidates for isotope conjugation. The aim of our preclinical study was to assess the biodistribution and tumor-targeting potential of ^52^Mn-labeled DOTAGA-bevacizumab. The VEGF-A targeting potential of the new immuno-PET ligand was assessed by using the VEGF-A expressing KB-3-1 (human cervix carcinoma) tumor-bearing CB17 SCID mouse model and in vivo PET/MRI imaging. Due to the high and specific accumulation found in the subcutaneously located experimental cervix carcinoma tumors, [^52^Mn]Mn-DOTAGA-bevacizumab is a promising PET probe for the detection of VEGF-A positive gynecological tumors, for patient selection, and monitoring the efficacy of therapies targeting angiogenesis.

## 1. Introduction

Cervical carcinoma is the fourth most common gynecologic malignancy in women behind, breast, colorectal, and lung cancer. In 2018, more than 500,000 women were diagnosed worldwide with cervical cancer, and of this high number, unfortunately, we have lost more than 300,000 patients with an incidence of 13.1/100,000 [1,2]. Improving the overall five-year survival rate, which was only 66% between 2010 and 2016, is essential [3].

Tumor-associated angiogenesis is a great concern, and it can immensely contribute to the progression and mortality of the disease. One of the players in tumor angiogenesis is the vascular endothelial growth factor (VEGF) family [4]. Different subtypes of VEGF exist (VEGF-A, -B, -C, -D, -E, and PDGF) from which VEGF-A is responsible for blood vessel growth. Its physiological role is to contribute to embryological, reproductive, and bone angiogenesis. VEGF-A can be overexpressed by several types of malignant cells. After secretion, VEGF can bind to VEGF receptors (VEGFR) located on the surface of endothelial cells. VEGF-A has a higher affinity to tyrosine kinase receptors such as VEGFR-1 and VEGFR-2. Mainly through VEGFR-2, signaling pathways can become activated (e.g., PI3K-Akt and PLCγ-PKC-MAPK pathways) that lead to both tumor growth and angiogenesis [5].

Bevacizumab, as an anti-VEGF antibody, is a widely used monoclonal antibody in the therapy of advanced cervical cancer [6]. It can block the proliferation and angiogenesis of cancer cells and, therefore, slow the progression of cervical carcinoma by binding to VEGF [7]. However, no biomarker has been developed that would be a useful modality to predict the effectiveness of bevacizumab therapy and monitor later-developed drug resistance.

Nuclear medicine plays an important role in diagnosing and staging malignancies and in treatment planning using ^18^F-FDG PET/CT [8]. Molecular imaging has further potential through radiolabeling targeted therapeutic agents, such as bevacizumab (immuno-PET) [9]. Within the tools of nuclear medicine and molecular imaging, PET/CT and PET/MRI—as hybrid imaging techniques—make visual assessment possible; moreover, the appearance of each lesion can be described objectively using numerical parameters obtained from the images [10].

Monoclonal antibodies are promising candidates for molecular imaging, as they specifically bind to their target molecule, and their labeling with a radioactive isotope can be easily carried out. Antibodies have a long circulation time in the blood, and due to this property, radioactive metal ions with a long half-life are the most suitable for their radioisotope labeling; however, metal ions are not capable of direct binding to immunoglobulins, so the use of chelating agents is necessary [9]. It is difficult to find an isotope that forms a stable metal ion-chelator complex. Several isotopes (e.g., ^89^Zr, ^86^Y, ^64^Cu, ^111^In) conjugated with bevacizumab have been investigated, from which zirconium-89 (^89^Zr) positron-emitting isotope seemed to be the most promising due to its ideal half-life (t_1/2_ = 3.27 days) [11]. A chelator suitable for the coordination sphere of ^89^Zr, which can prevent its release from the complex leading to the accumulation of the isotope in the bones, has not been found so far [12]. According to this, the use of an isotope whose stable complex can be more easily produced would be a safer and better choice for patients. Manganese-52 (^52^Mn) (t_1/2_ = 5.59 days) is a novel positron-emitting isotope, so there is not much literature about its use as a PET radiotracer, but a lot of data on the complex-forming properties of Mn^2+^ ion have been collected due to the use of manganese as an MRI contrast agent [13,14,15,16].

The aim of this preclinical study was to investigate the in vivo biodistribution and tumor targeting potential of the newly synthesized ^52^Mn-labeled DOTAGA-bevacizumab PET probe in the VEGF-A positive cervix carcinoma tumor-bearing mouse model.

## 2. Materials and Methods

### 2.1. General

For the radioactivity measurements, a Perkin Elmer Wizard gamma counter and the MED Isomed 2010 dose calibrator were used. To confirm the success of the labeling reactions, a radio-HPLC was used with the Waters Acquity UPLC I-class system, connected to a radioactivity detector (Berthold LB513; Radchem Co. Ltd., Budapest, Hungary) with a 20 µL plastic scintillator (MX) cell.

### 2.2. Chemicals

Rotipuran Ultra H_2_O (u.p. H_2_O), 34% Rotipuran Ultra HCl (u.p. HCl), and Pufferan ≥ 99.5% Cellpure HEPES were purchased from Carl Roth. Ultra-pure ammonium acetate (u.p. NH_4_OAc) was bought from VWR. HPLC-MS grade ACN was provided by Scharlau. The Xbridge Premier Protein SEC column (250 Å 2.5 µm, 4.6 × 150 mm) was supplied by Waters.

### 2.3. Radiolabeling

Bevacizumab (Avastin^®^; Roche Pharma AG, Grenzach-Wyhlen, Germany) was derivatized with a fivefold excess of 2,2′,2″-(10-(1-carboxy-4-((4-isothiocyanatobenzyl)amino)-4-oxobutyl)-1,4,7,10-tetraazacyclododecane-1,4,7-triyl)triacetic acid (p-NCS-Bn-DOTA-GA; Chematech) in NaHCO_3_ buffer (pH 8.2) at room temperature. DOTAGA-bevacizumab was purified by ultrafiltration on an Amicon Ultra (Merck KGaA, Darmstadt, Germany), 0.5 mL, 30 kDa centrifugal filter (Millipore). Radiolabeling was implemented at pH 6 for 15 min at room temperature. The required pH value was adjusted by sodium-acetate (0.04 M) buffer and sodium hydroxy. The efficiency of radiolabeling was followed by Raytest miniGita Star radio-thin layer chromatography scanner. 3 µL of the reaction mixture was dropped onto a glass macrofibre chromatography paper impregnated with silica gel (iTLC-SG) strip and developed in 0.1 M sodium citrate solution. [^52^Mn]Mn-DOTAGA-bevacizumab was remained near the start point (R_f_ = 0.1–0.2), whereas ^52^Mn^II^ as citrate complex was eluted with the solvent front (R_f_ = 0.8–1). The [^52^Mn]Mn-DOTAGA-bevacizumab product was used without further purification in experiments.

### 2.4. Determination of In Vitro Stability of [^52^Mn]Mn-DOTAGA-Bevacizumab

[^52^Mn]Mn-DOTAGA-bevacizumab was diluted with fourfold excess of 0.01 M EDTA (pH 7.3), 0.01 M oxalic acid (pH 5.7), and mouse serum solution to investigate its stability. In the case of stability experiments against EDTA and oxalix acid, the reaction was carried out at 25 °C for 0.17, 1.5, 3, 21, and 47 h. In the case of stability investigation in mouse serum, the reaction mixture was incubated at 37 °C, and sampling was performed at 0.2, 4, 22, 47, 118, and 167 h, where samples were diluted fivefold with water before directly injected into UPLC. In each case, samples were analyzed on an Xbridge Premier Protein SEC 250 column (Waters) using an isocratic method with 0.45 mL/min flow rate, where the liquid phase was 100 mM ammonium acetate (pH 7.2) solution.

### 2.5. Cell Lines

Human KB-3-1 cervix carcinoma cell line [17] was obtained from Dr. Katalin Goda (University of Debrecen, Faculty of Medicine, Department of Biophysics and Cell Biology). For cell culturing, the Dulbecco’s Modified Eagle’s medium (DMEM, GIBCO Life Technologies Magyarország Ltd., Budapest, Hungary) was used supplemented with Fetal Bovine Serum (10%, heat-inactivated FBS from GIBCO, Life technologies Magyarország Ltd., Budapest, Hungary) and Antibiotic and Antimycotic solution (1%, Sigma-Aldrich, Merck KGaA, Darmstadt, Germany). KB-3-1 cells were maintained at standard culturing conditions (5% CO_2_ and 37 °C). For subcutaneous tumor inoculation, cervix carcinoma cells were used at 80% confluency, and the cell viability was confirmed by the trypan blue exclusion test.

### 2.6. In Vivo Cervix Carcinoma Tumor Model

Immunodeficient CB17 SCID mice (12-week-old; female; *n* = 35) were housed under sterile conditions in individually ventilated cage system (IVC cages, Techniplast, Akronom Ltd., Budapest, Hungary) under standard conditions (25 ± 2 °C and 55 ± 10%). A sterile semi-synthetic rodent diet (SDS VRF, Animalab Ltd., Budapest, Hungary) and sterile tap water were available ad libitum to all the experimental animals. Experimental animals were kept and treated in accordance with all the corresponding paragraphs of the Hungarian Ethical Laws and the regulations of the European Union (permission number: 16/2020/DEMÁB).

For the establishment of the KB-3-1 cervix carcinoma tumor model, experimental animals were anesthetized by 3% isoflurane (Forane, AbbVie, Budapest, Hungary; OGYI-T-1414/01), O_2_ 0.4 L/min and N_2_O 1.2 L/min (Linde Healthcare, Budapest, Hungary; OGYI-T-20607 and OGYI-T-21090, respectively) using the Tec3 Isoflurane Vaporizer anesthesia device (Eickemeyer, Sunbury-on-Thames, Surrey, UK), then 5 × 10^6^ KB-3-1 tumor cells in saline (150 µL 0.9% NaCl) were injected subcutaneously into the right shoulder area of the experimental animals. In vivo imaging and ex vivo biodistribution studies were carried out 11 ± 1 days after the implantation of KB-3-1 cancer cells at the tumor volume of approximately 75 mm^3^.

### 2.7. In Vivo PET/MRI Imaging

KB-3-1 cervix carcinoma tumor-bearing mice were anesthetized by 1.5% isoflurane, and for the anatomical localization of the investigated tissues, whole-body MRI scans (T1-weighted) were performed using the preclinical *nanoScan* PET/MRI system (Mediso Ltd., Budapest, Hungary). The 3D GRE EXT multi-FOV MRI parameters were set as follows: TR/TE 15/2 ms; phase: 100; FOV 60 mm; NEX 2. After MRI imaging, animals were injected intravenously with 9.43 ± 1.03 MBq of ^52^MnCl_2_, [^52^Mn]Mn-DOTAGA, or [^52^Mn]Mn-DOTAGA-bevacizumab and dynamic PET scans were performed. The co-registered and reconstructed (3D-OSEM algorithm with Tera-Tomo reconstruction software, Mediso Ltd., Budapest, Hungary) decay-corrected PET images were analyzed by the InterView™ FUSION (Mediso Ltd., Budapest, Hungary) image analysis software. Volumes of Interest (ellipsoidal 3-dimensional VOIs) were manually drawn around the edge of the activity of the investigated tissues and organs by visual inspection. The accumulation of the ^52^Mn-labeled probes was expressed in terms of standardized uptake values (SUVs).

### 2.8. Immunohistochemistry

For the detection of the VEGF-A expression of the subcutaneously growing KB-3-1 cervix carcinoma, xenografts cryosections (5 µm thick) were made from the tumors. Sections were dried, fixed (10 min at −20 °C in pre-cooled acetone), blocked (20 min with 1% BSA-PBS), and were further incubated at 24 °C with anti-human VEGF Alexa Fluor^®^ 488-conjugated monoclonal antibody (IC2931G; Bio-Techne R&D Systems Ltd. Budapest, Hungary). For nuclear counterstaining DAPI, (MBD0020; DAPI ready-made solution with Antifade; Merck, Darmstadt, Germany) was used. For fluorescence imaging, the Zeiss LSM 510 confocal laser-scanning microscope was used.

### 2.9. Statistical Analysis

The statistical significancy was assessed by the Student’s t-test (two-tailed), two-way ANOVA, and Mann-Whitney U-test using the MedCalc software (MedCalc Software v18.5., Mariakerke, Belgium; https://www.medcalc.org, accessed on 17 December 2022). Quantitative data were presented as mean ± SD, and the level of significance was set at *p* < 0.05.

## 3. Results

### 3.1. Radiolabeling and Characterization of [^52^Mn]Mn-DOTAGA-Bevacizumab

The chemical purity of the DOTAGA-bevacizumab precursor was verified with an HPLC-UV-MS system. The DOTAGA/bevacizumab ratio was calculated based on the consumption of DOTAGA pSCN-Bn, during which the DOTAGA/bevacizumab ratio was 4.42 ± 0.26. DOTAGA-bevacizumab was labeled with ^52^MnCl_2_ (Figure 1) with a specific activity of 0.01 MBq/μg. The final radiochemical yield (RCY) was >90%, measured by thin-layer chromatography. The stability of [^52^Mn]Mn-DOTAGA-bevacizumab was investigated in EDTA, oxalic acid solution, and mouse serum and assessed by size-exclusion chromatography. [^52^Mn]Mn-DOTAGA-bevacizumab was stable against EDTA and oxalic acid, with only about a 5% decrease in radiochemical yield over 48 h (Figure 2A). The radiochemical yield of [^52^Mn]Mn-DOTAGA-bevacizumab was above >70% for up to 7 days in mouse serum (Figure 2B).

### 3.2. Biodistribution and PET/MRI Imaging Studies

For the assessment of VEGF receptor specificity of [^52^Mn]Mn-DOTAGA-bevacizumab, in vivo tissue distribution studies were executed by healthy and KB-3-1 tumor-bearing SCID mice. In the first experiments, for the confirmation that the newly synthesized [^52^Mn]Mn-DOTAGA-bevacizumab probe is stable in vivo, the uptake values of the biodistribution were compared to that of ^52^MnCl_2_ (Figure 3) and [^52^Mn]Mn-DOTAGA in healthy control mice (Figure 4). The SUV_mean_ values of the selected organs were assessed at seven different time points (4 h, 1, 2, 3, 5, 7, and 10 days) after tracer injection. The qualitative PET image and the quantitative SUV data analysis showed that the accumulation of ^52^MnCl_2_ in the liver and kidney cortex was initially remarkable; however, the ^52^MnCl_2_ content in these organs decreased with time. The activity concentration of the pancreas and salivary glands also showed high values, which did not decrease as time progressed (Figure 3).

In the case of [^52^Mn]Mn-DOTAGA, only the uptake of the lungs was prominent; however, this was also decreased rapidly, and [^52^Mn]Mn-DOTAGA showed very low uptake values in all of the other investigated organs due to the rapid clearance at the early time points (Figure 4). [^52^Mn]Mn-DOTAGA-bevacizumab showed elevated uptake in the blood, liver, kidney, spleen, and lung, although a continuous decrease in activity concentration was also observed in these organs till ten days post-injection (Figure 5). In summary, [^52^Mn]Mn-DOTAGA showed significantly lower uptake in all examined organs than that of the two other radiotracers.

The biodistribution and VEGF-A receptor specificity of the newly synthesized [^52^Mn]Mn-DOTAGA-bevacizumab was assessed by preclinical PET/MRI imaging using KB-3-1 tumor-bearing mice. After the qualitative image analysis, it was found that the subcutaneously growing KB-3-1 cervix tumors were identifiable from 4 h after the tracer injection; moreover, an increasing accumulation of radiopharmaceuticals was observed as time progressed (Figure 5A). The quantitative SUV analysis of the PET images showed that the accumulation of [^52^Mn]Mn-DOTAGA-bevacizumab was increasing until day 2–3 (SUV_mean_: approx. 2) in the KB-3-1 tumors, then a decrease was observed with slow kinetics; however, the accumulation remained high for the rest of the investigated time points (SUV_mean_: approx. 1.2 at day ten post-injection) (Figure 5B, insert). The presence of VEGF-A in the examined tumors was confirmed by immunohistochemical staining performed on day ten post injection of [^52^Mn]Mn-DOTAGA-bevacizumab, and a strong expression was found in the membrane of the cancer cells (Figure 5C) confirming the target-specific property of the radiopharmaceutical.

Tumor-to-organ ratios (SUV_mean_ tumor/SUV_mean_ organ) were assessed by quantitative SUV data analysis of the PET images obtained from KB-3-1 tumorous mice after the i.v. injection of [^52^Mn]Mn-DOTAGA-bevacizumab. It was generally observed that the ratio of the SUV_mean_ values started to plateau 2–3 days after ^52^Mn-labeled DOTAGA-bevacizumab was injected, and these ratios remained high until the end of the study (ten days post-injection); however, in some cases (tumor-to-liver, -blood, -spleen) a slight increase was observed in the SUV_mean_ ratios from day three to day ten post-injection of [^52^Mn]Mn-DOTAGA-bevacizumab (Figure 6).

## 4. Discussion

Among humanized monoclonal antibodies, bevacizumab specifically binds to vascular endothelial growth factor A (VEGF-A), which is an overexpressed biomarker in different tumor types and plays an important role in the development and maintenance of tumor-associated angiogenesis. The standard protocol for biomarker quantification is generally biopsy sampling and immunohistochemistry (IHC) or mRNA validation. Nevertheless, the evaluation of biomarker expression is influenced by the tumor heterogeneity and sampling errors. In addition, repetitive biopsies and histopathological confirmation are required to monitor treatment response, making clinical use challenging [18,19]. In contrast, the nanomolar sensitivity of the non-invasive positron emission tomography using radiolabeled target-specific monoclonal antibodies (immuno-PET) provides the longitudinal and quantitative assessment of tumor target expression [20,21].

In this present study, the VEGF-A targeting ability of the new immuno-PET probe was assessed by using the VEGF-A expressing KB-3-1 (human cervix carcinoma) tumor-bearing CB17 SCID mouse model and in vivo PET/MRI imaging. The new [^52^Mn]Mn-DOTAGA-bevacizumab PET probe was synthesized with high radiochemical purity and appropriate stability properties (Figure 1 and Figure 2). Similar stability data were also described by other authors (e.g., ^99m^Tc-labeled BevMab-DTPA, ¹³¹I-bevacizumab); however, they examined the stability in serum for 24 and 48 h, respectively, due to the difference in the half-life of the radionuclides used [22,23].

By the biodistribution studies, it has been found that the fastest clearance was seen with [^52^Mn]Mn-DOTAGA through the kidney, and therefore, its activity was very low at all the investigated time points (Figure 4). ^52^MnCl_2_ (Figure 3) and [^52^Mn]Mn-DOTAGA-bevacizumab needed longer time for excretion. ^52^Mn^2+^ was excreted mainly via the kidneys, while the antibody stayed in the blood for a long time and showed relatively high uptake in the lung (Figure 5).

Bevacizumab is specific for the human VEGF and inhibits its biological activity [24]. It is also known that the presence and over-expression of VEGF in cervix carcinomas is an outstanding prognostic marker and is associated with poor patient survival [25,26]. Accordingly, we also found strong positivity for VEGF expression by immunohistochemical studies and found high and specific accumulation of [^52^Mn]Mn-DOTAGA-bevacizumab in the VEGF-A positive KB-3-1 cervical cancer xenografts (Figure 5), however. In addition, an important parameter for the assessment of PET images is the radioactivity concentration of healthy organs and tissues compared to the tumor. These data also provide information on non-specific and off-target binding. The evaluation of these data is also important due to it was observed that bevacizumab binds with high affinity to, for example, brain receptors (dopamine, GABA, histamine), whereas the amounts of substances used in PET diagnostics do not cause a pharmacological effect or side-effect [24]. In this present study, the tumor-to-background ratios increased and reached a plateau 2–3 days after the administration of ^52^Mn-labeled bevacizumab and remained high until the end of the study, indicating the specific accumulation and high binding affinity in the tumors (Figure 6).

From our observations, we can conclude that [^52^Mn]Mn-DOTAGA-bevacizumab could be a useful tracer for patient selection to bevacizumab therapy, as well as monitoring the efficacy of the targeted therapy and later developed drug resistance. Numerous preclinical studies have been published using different isotopes for this purpose, including ^116^Ho, ^89^Zr, ^111^In, ^86^Y, and ^99m^Tc [27,28,29,30,31]. Furthermore, several human clinical studies are known from the literature in which ^89^Zr-labeled bevacizumab has outstanding tumor targeting and PET imaging ability [32,33,34,35,36,37]. However, the above-mentioned isotopes have shorter half-lives than ^52^Mn. Since bevacizumab stays in the circulation for a long time, and tumor-to-background ratios remain elevated for several days, ^52^Mn may potentially be a more ideal candidate to use for immunoPET imaging of anti-VEGF target drug than isotopes with shorter half-lives. Moreover, the favorable and well-described chelating properties of Mn^2+^ make it possible to use chelators that form stable complexes not only with ^52^Mn but also with β^–^ and α emitting therapeutic isotopes like ^177^Lu and ^225^Ac [38,39]. This feature could lead to the use of bevacizumab as a theranostic agent as well.

## 5. Conclusions

In conclusion, due to high and specific accumulation observed in the subcutaneously growing KB-3-1 experimental cervix carcinoma tumors, [^52^Mn]Mn-DOTAGA-bevacizumab is a promising radiopharmaceutical in the imaging of VEGF-A positive gynecological tumors, in patient selection, and monitoring the efficacy of therapies targeting angiogenesis.

## Figures and Tables

**Figure 1 diagnostics-13-00236-f001:**
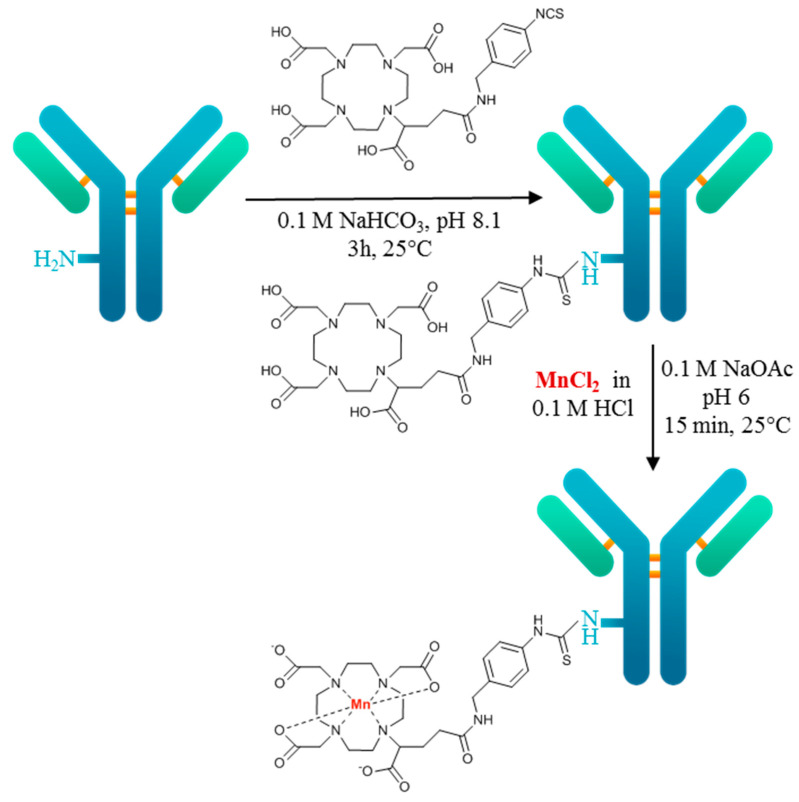
Scheme of the radiochemical synthesis of [^52^Mn]Mn-DOTAGA-bevacizumab.

**Figure 2 diagnostics-13-00236-f002:**
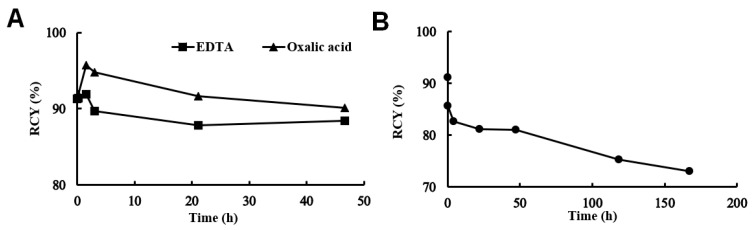
Stability of [^52^Mn]Mn-DOTAGA-bevacizumab in EDTA and oxalic acid (**A**) and mouse serum (**B**).

**Figure 3 diagnostics-13-00236-f003:**
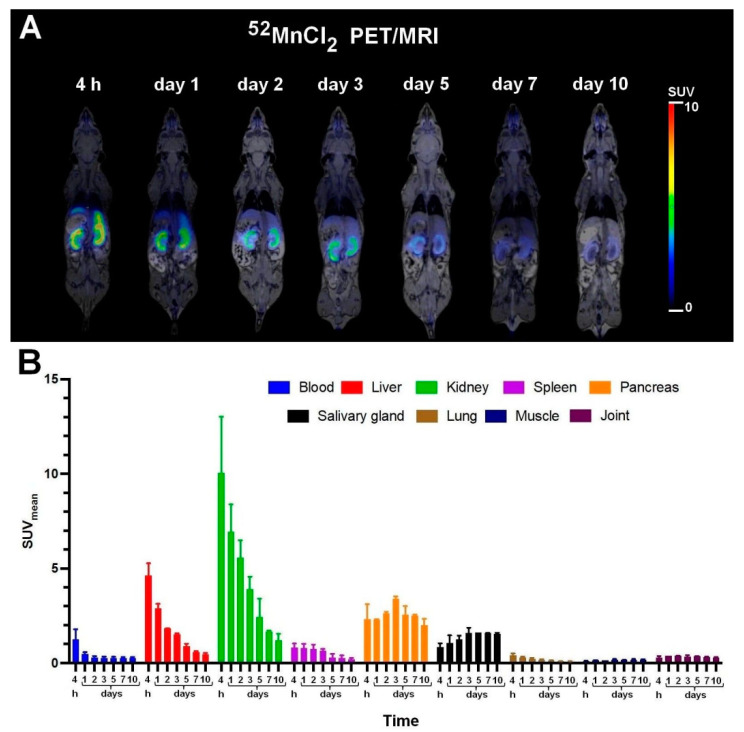
In vivo PET/MRI imaging of ^52^MnCl_2_ biodistribution (**A**) and average time-activity changes (**B**) of the selected organs 4 h, 1, 2, 3, 5, 7, and 10 days after the intravenous injection of approximately 10 MBq of ^52^MnCl_2_. SUV: standardized uptake value. Data are shown as mean ± SD and obtained from *n* = 5 animals.

**Figure 4 diagnostics-13-00236-f004:**
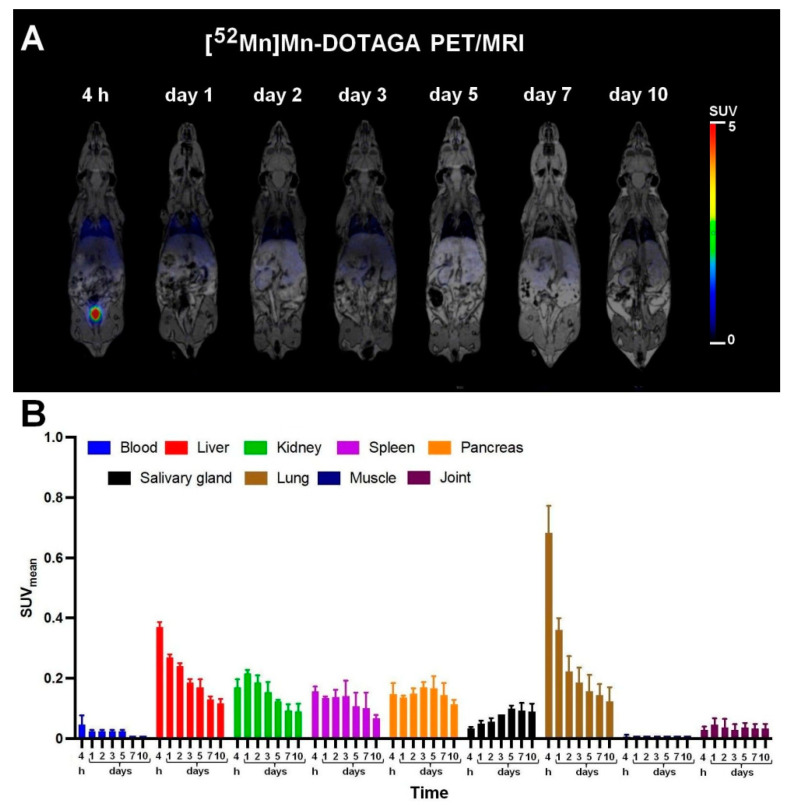
In vivo PET/MRI imaging of [^52^Mn]Mn-DOTAGA biodistribution (**A**) and average time-activity changes (**B**) of the selected organs 4 h, 1, 2, 3, 5, 7, and 10 days after the intravenous injection of approximately 10 MBq of [^52^Mn]Mn-DOTAGA. SUV: standardized uptake value. Data are shown as mean ± SD and obtained from *n* = 5 animals.

**Figure 5 diagnostics-13-00236-f005:**
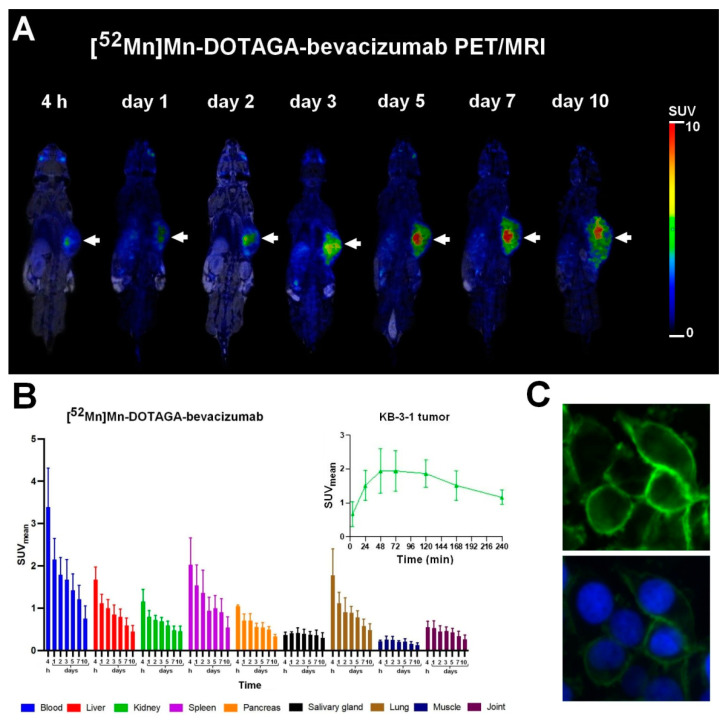
In vivo PET/MRI imaging of KB-3-1 cervix carcinoma tumor-bearing mice (**A**). Representative decay-corrected coronal PET/MRI images were obtained from 4 h to 10 days post injection of [^52^Mn]Mn-DOTAGA-bevacizumab. White arrows: subcutaneously growing KB-3-1 cervix tumors. (**B**) The panel shows the biodistribution and the average time-activity curve (TAC) for the VEGF-A positive KB-3-1 tumors 4 h, 1, 2, 3, 5, 7, and 10 days after the intravenous injection of approximately 10 MBq of [^52^Mn]Mn-DOTAGA-bevacizumab. SUV: standardized uptake value. Data are presented as mean ± SD and obtained from *n* = 5 animals/time point. (**C**) panel demonstrates the VEGF-A positivity of subcutaneously growing KB-3-1 tumor by immunohistochemical study. Upper image: VEGF-A receptors (green color by Alexa-488 staining); lower image: VEGF-A receptors and nuclear counterstaining (blue color by DAPI staining).

**Figure 6 diagnostics-13-00236-f006:**
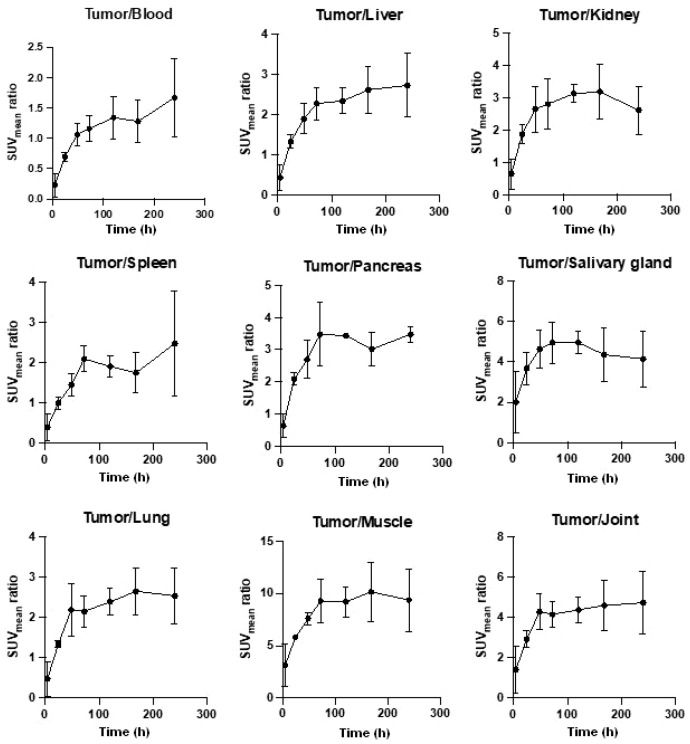
Quantitative assessment of KB-3-1 tumor-to-organ ratios in CB17 SCID mice after the i.v. injection of [^52^Mn]Mn-DOTAGA-bevacizumab. SUV: standardized uptake value. Data are shown as mean ± SD and obtained from *n* = 5 animals/time point.

## Data Availability

The dataset used and analyzed in the current study are available from the corresponding author on reasonable request.

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
