# Peer review of "In Vivo Preclinical Assessment of the VEGF Targeting Potential of the Newly Synthesized [52Mn]Mn-DOTAGA-Bevacizumab Using Experimental Cervix Carcinoma Mouse Model"

_diagnostics, 2023, doi:10.3390/diagnostics13020236_

Round 1
Reviewer 1 Report
The paper of Trencsényi and collaborators reports the preparation and in vivo preclinical assessment of [52Mn]Mn-DOTAGA-bevacizumab on subcutaneously injected KB-3-1 tumor-bearing mice. Bevacizumab targets VEGF-A and it is used in the antiangiogenesis therapy, therefore the authors propone the use of this PET ligand for patient selection to bevacizumab therapy or to monitor the effectiveness of bevacizumab therapy (or other therapies targeting angiogenesis).
Since the monoclonal antibodies have a long circulating time, the authors suggest that the use of manganese-52 positron-emitting isotope could be ideal because of its long half-life (t1/2= 5.59 days). 52Mn is an interesting emerging radiometal for immuno-PET, and there are only few studies reporting on the in vivo evaluation of a 52Mn-labeled mAb, and for tis reason this paper is an interesting contribution to this field.
The results are good in terms of tumor accumulation and tumor-to-organ ratios and well-described. For these reasons I suggest the publication in Diagnostics.
Only a point should be clarified. The authors describe the synthesis of DOTAGA-Bevacizumab, and at line 270, the say that “the new [52Mn]Mn-DOTAGA-bevacizumab PET probe was synthesized with high chemical purity”. How did they check the chemical purity? Did they perform a mass analysis? How many DOTAGA units were bound per antibody? These data should be interesting for a reader.
Other minor points:
At Line 94 the authors write they used Trastuzumab (Ontruzant®), and so at line 97 DOTAGA-trastuzumab. Please correct.
Some typos:
Line 128 Immunodefficient
Line 247 that he ratio
Line 296 the administration if
Reviewer 2 Report
Very nice research and the manuscript.
It would be nice if you could cite some clinical experience or clinical trials with MaB and radiopharmaceuticals in oncology.
Reviewer 3 Report
Dear Author,
Please find the minor comments.
Rewrite second sentence in introduction.
41st line - pdgf is correct not PIGE
Rewrite 45th sentence.
comma is missing in 299 prepositional phrase.
`
